# Spatio-Temporal Variation-Induced Group Disparity of Intra-Urban NO_2_ Exposure

**DOI:** 10.3390/ijerph19105872

**Published:** 2022-05-12

**Authors:** Huizi Wang, Xiao Luo, Chao Liu, Qingyan Fu, Min Yi

**Affiliations:** 1Key Laboratory of Road and Traffic Engineering of the Ministry of Education, College of Transportation Engineering, Tongji University, Shanghai 201804, China; wanghuizi@tongji.edu.cn; 2College of Architecture and Urban Planning, Tongji University, Shanghai 200092, China; 3Shanghai Tongji Urban Planning and Design Institute Co., Ltd., Shanghai 200092, China; 4Shanghai Environmental Monitoring Center, Shanghai 200235, China; qingyanf@sheemc.cn (Q.F.); yimin@sheemc.cn (M.Y.)

**Keywords:** land use regression (LUR), mobile phone signal data, air pollution, population exposure, environmental justice

## Abstract

Previous studies on exposure disparity have focused more on spatial variation but ignored the temporal variation of air pollution; thus, it is necessary to explore group disparity in terms of spatio-temporal variation to assist policy-making regarding public health. This study employed the dynamic land use regression (LUR) model and mobile phone signal data to illustrate the variation features of group disparity in Shanghai. The results showed that NO_2_ exposure followed a bimodal, diurnal variation pattern and remained at a high level on weekdays but decreased on weekends. The most critical at-risk areas were within the central city in areas with a high population density. Moreover, women and the elderly proved to be more exposed to NO_2_ pollution in Shanghai. Furthermore, the results of this study showed that it is vital to focus on land-use planning, transportation improvement programs, and population agglomeration to attenuate exposure inequality.

## 1. Introduction

Air pollution has become one of the most serious environmental problems globally. Worldwide, approximately 4.2 million premature deaths could be attributed to air pollution in 2016 [1]. In China, 112.7 of every 100,000 deaths were caused by air pollutants [2]. The rapid development of heavy chemical industries, energy consumption, and motor vehicle ownership has resulted in massive emissions of nitrogen dioxide (NO_2_), which have led to increasing pollution by secondary pollutants such as acid rain and photochemical pollution. Epidemiological investigations have confirmed that NO_2_ pollution is associated with considerable health risks, even below the current WHO air quality guidelines [3]. Short-term exposure to NO_2_ pollution may cause airway responsiveness and lung function injury, while long-term exposure may impair the immune function, even potentially increasing the risk and severity of respiratory viral infection [4,5,6]. Furthermore, mounting evidence indicates that vulnerable social groups are more likely to be exposed to air pollution [7,8]. However, most studies on exposure inequality focused on the annual average data, ignoring the impacts of temporal variation of air pollution. Therefore, it is necessary to conduct studies on the group disparity in population exposure assessment in terms of spatio-temporal variation to assist policy-making aimed at attenuating exposure inequality and improving public health.

Exposure is defined by both air pollution concentration in places where people spend time and the amount of time spent in each place [9,10]. It relies on both air pollutant concentration and population distribution. The land use regression (LUR) model is recognized as a common method to simulate spatial–temporal variations in air pollution. Most studies use static explanatory variables and static data, illustrating a strong spatial but poor temporal variation of air pollution. This study introduced hourly dynamic data to delineate the high-resolution temporal heterogeneity of air pollution. Furthermore, mobile phone signal data were employed to measure population dynamics, which helped to improve the accuracy of the exposure assessment. We proposed a case study of Shanghai to assess NO_2_ exposure and the spatio–temporal variation and group disparity (age or gender) in Shanghai based on a dynamic LUR model and mobile phone signal data.

In summary, the main contributions of this study were two-fold: (1) a fairly novel approach that integrated a dynamic LUR model and mobile phone signal data was employed to improve the accuracy of dynamic population exposure; (2) a more detailed analysis of group disparity was investigated in terms of spatio-temporal variation.

The remainder of this paper is structured as follows: Section 2 reviews the LUR application to the spatio-temporal variation in air pollution, analyzes the development of air pollution exposure, and discusses exposure equality and environmental justice. Section 3 describes our model framework regarding NO_2_ exposure built with a dynamic LUR model and mobile phone signal data. Section 4 presents the results of our model and discusses areas of future interest for spatio-temporal variation and group disparity in air pollution exposure. Section 5 discusses the principal findings, strengths, and limitations of the study and policy implications. Section 6 concludes the paper.

## 2. Literature Review

### 2.1. LUR Application in Spatio-Temporal Variations of Air Pollution

The LUR model has been proven to be a valid and cost-effective approach for assessing exposure to air pollutants in epidemiological investigations, especially for traffic-generated air pollution [11]. Over the past few years, it has been applied extensively in European [12,13,14] and North American settings [15,16,17,18,19] and has gradually become popular in China [20,21,22,23]. 

Focusing on static data leads to a LUR model with superior spatial but poor temporal variations [24]. Therefore, highlighting the spatio-temporal variation is recognized as a focus and orientation for research. Ever more attention is being focused on improving the sensitivity of the LUR model to the temporal variation, and several methods have been applied. The first method is to recalibrate the intercept of LUR models with the temporal difference of pollutant concentrations [25,26]. The second method is to introduce dummy time variables to the LUR models to represent different periods [24,27,28]. The third method is to apply dynamic variables to establish several models in different periods [24,29,30]. In addition, earlier studies have combined the above methods to model the spatial-temporal variations of air pollutants [31,32]. Recently, the spatio-temporal LUR model has provided a more in-depth understanding of the dynamic variation of air pollution and the relationship between pollutant concentrations and the urban environment, but it remains limited because these models are mainly focused on annual [33,34] or seasonal variations [35,36]. Delineating high-resolution temporal heterogeneity (hourly, diurnal, weekly, monthly, etc.) remains difficult, making medium- and short-term exposure studies non-feasible. Thus, an in-depth understanding of high-resolution temporal variation is crucial for estimating exposure to air pollution accurately.

### 2.2. Air Pollution Exposure Assessment

Accurate exposure-based health risk assessments require not only high-resolution measurements of air quality but also knowledge of human activity patterns [37,38,39]. Earlier studies have typically focused more on static population data such as census data [40,41,42,43,44,45,46] and nighttime lighting [47,48]. However, evidence has shown that exposure levels are generally underestimated by static analysis, as it ignores variability in exposure caused by individual mobility [49,50,51,52,53,54,55]. Recent studies have attempted to introduce travel surveys [56,57], activity-based dynamic population models [49,58,59,60], global positioning system (GPS) models [53,61,62,63], travel smartcards [64], call detail records (CDRs) [65], or location-based service (LBS) data [66,67,68] to analyze dynamic exposure levels. These approaches provide more detailed spatio-temporal analyses of individual travel behavior, but, for researchers and participants, these methods are often inefficient, expensive, and limited in terms of how many people they can track. 

Today, increasingly available mobile phone signal data provide a new opportunity to further improve measurements of population exposure. However, to date, limited efforts have been made to use large-scale mobile phone signal data to examine variations in exposure to air pollution [51,69]. The first study that used mobile phone data to examine the disparity in exposure to air pollution in China was performed by Guo et al. in 2020 [70]. This study utilized data collected on a weekday (23 March 2012) in Shenzhen. Given the limited time frame, medium- and short-term population exposure remains unclear. Furthermore, this study inappropriately applied time-profile population exposure to assess individual-level exposure over a large spatial area instead of the microenvironment such as an individual’s residence or workplace [71].

### 2.3. Exposure Equality and Environmental Justice

Environmental justice believes that regardless of socioeconomic status, all residents ought to enjoy the equal benefits of public natural resources and shoulder equal adverse health impacts from deteriorating environmental conditions [72,73,74]. Increasingly, the literature has been illustrating that several subpopulations may have born a disproportionate air pollution exposure and health burden over the past several decades [75]. This is of particular concern because a better understanding of the impacts of air pollution exposure on environmental justice is a public health policy-making priority. However, there exists a gap in China due to the lack of data and neglect for environmental justice issues.

Many environmental justice studies have noted the inequitable distribution of air pollution exposure among marginal social groups (e.g., poor people, Blacks, and children) [7,8], in part because these vulnerable populations are more likely to reside near air pollutant sources where cheaper housing and more job opportunities congregate. For example, a national study in the United States has revealed that low-income non-White children and the elderly were more likely to be exposed to NO_2_ [76]. A similar conclusion was drawn in Sweden, where children living in poorer conditions were burdened with higher NO_2_ exposure [77]. Race or ethnicity is a particularly consequential element when it comes to air pollution exposure disparity. Many studies in the USA have demonstrated that Blacks and Latinos are disproportionately exposed to higher levels of NO_2_ than whites [46,78,79,80]. Evidence from nine European metropolitan areas, indicates that higher NO_2_ values were observed in areas with higher populations of individuals born outside the European Union [81]. However, some cases contradict the above studies; for example, Toronto reported that racial minority groups tended to have less air pollution exposure, probably due to the immigration policy aimed at highly educated groups [82]. Low-income populations also suffer more adverse impacts of air pollution, even they have contributed little to the pollution [83,84,85]. 

In addition, recent environmental justice research attempts to understand the evolution of environmental inequality through long-term dynamic analysis. These studies found that vulnerable populations benefit the least from air-quality improvement; in other words, exposure decreased more in less polluted areas, which means that group disparity in air pollution increased further [86,87,88].

## 3. Methodology

In this section, the study area and research data we used to model aggregate-level NO_2_ exposure are introduced in detail, followed by the modeling and validation of the LUR model and the calculation and comparison of population exposure. The research flow chart is shown in Figure 1.

### 3.1. Study Area

Shanghai is one of the municipalities under direct administration of the Central Government of China and at the core of the world-class city cluster in the Yangtze River Delta area. It is composed of 16 districts with an area of 6340.5 km^2^ and a total population of 24.28 million as of the end of 2019. There were 56 days when air pollution was severe during 2019 [89], and the primary pollutants were fine particles and NO_2_. The mean value of NO_2_ concentration was 42 μg/m^3^ in Shanghai, 2 μg/m^3^ higher than the standard set by the WHO (40 μg/m^3^) [90]. In general, severe air pollution still hangs over Shanghai. 

It should be mentioned that the study area in this study includes all districts except the isolated Chongming Island due to the differences in the geographical location and climate conditions from other districts as shown in Figure 2.

### 3.2. LUR Model Setting

A spatio-temporal LUR model was developed by utilizing hourly dynamic variables to establish several models in different periods. Considering that road transport is one of the primary contributors to the ambient concentration of NO_2_, a day was divided into four periods, namely, morning peak (7 a.m.–9 a.m.), daytime (9 a.m.–5 p.m.), evening peak (5 p.m.–7 p.m.), and nighttime (7 p.m.–7 a.m.), regarding traffic states in Shanghai [91].

#### 3.2.1. Dependent Variable

The mean values of the hourly concentrations of NO_2_ during different periods for 50 monitors in the 16 districts were retrieved from the Shanghai Environmental Monitoring Center and cover a time span from 11 November to 30 November 2019.

#### 3.2.2. Independent Variables

The independent variables included location, meteorological elements, road network, land use, point of interest, and other pollutants as listed in Table 1. It should be mentioned that the suburban and urban areas were set as binary variables, with 1 referring to urban monitoring stations and 0 to suburban monitoring stations, for which the boundary was the outer-ring highway in Shanghai. Mean hourly data for meteorological elements were extracted from Inverse distance weight (IDW) interpolation pollution maps which were drawn in ArcGIS based on the meteorological data from China Meteorological Data Service Center. The road network was elaborated as the intensity of highways and local roads, whose maximum buffer distance of the road networks was set at 1000 m, following the principle proposed by Hock [92]. Land-use data contained five types, namely, residential land use, commercial land use, industrial land use, green space, and water body, which were calculated in each buffer zone from monitor stations. Four types of points of interest (POIs) were extracted using the Baidu Open Map Platform (2019) based on categories and keywords. The numbers of restaurants, bus stops, intersections, and gas stations were calculated within each buffer.

#### 3.2.3. Modeling and Validation

The explanatory variables were chosen by a supervised forward regression method [93,94]. First, each potential explanatory variable was assigned to a prior direction to enhance the applicability of the LUR models described in Table 1. For example, wind speed could improve the dilution and diffusion process of pollutants in the atmosphere; thus, its prior direction was assumed as negative (denoted as “−“). A stable atmospheric configuration under a high-pressure environment was not conducive to the physical diffusion of pollutants, leading to the accumulation of pollutant concentrations. Therefore, its prior direction was assigned as positive (denoted as “+”). As for relative humidity, some studies have shown that higher relative humidity is detrimental to the diffusion process of gaseous pollutants, worsening environmental pollution, but other studies came to the opposite conclusion that higher relative humidity improves the transition of gaseous pollutants to particulate state, reducing the pollution concentration. Consequently, the prior direction of relative humidity was not specified since it’s still a controversial correlation and denoted by “O”.

Secondly, univariate regression was carried out between the dependent variables and each independent variable, and the resulting F-statistic and corresponding *p*-value were used to determine the significance of the predictor and the order for its entry into the model. The model with the highest adjusted *R*^2^ that was consistent with the prior direction was selected as the initial model of stepwise regression. Thereafter, a supervised forward regression was conducted in IBM SPSS (version 26.0), and the variables were introduced into the model if they satisfied the following criteria: (1) the adjusted *R*^2^ of the model increased by at least 1%; (2) the coefficient of each variable was consistent with the prior direction; (3) the existing variables in the model did not change their effect directions. All potential explanatory variables were introduced into the model one by one until there were no remaining variables that satisfied the above criteria. Then, the variables with *p*-values greater than 0.1 were excluded from the model. 

Thirdly, standardized diagnostic tests were applied to the final models to check the multicollinearity between the variables and influential observations. The variables with the highest variance inflation factors (VIFs) of more than 3 were excluded from the final model and the model was recalibrated. 

Finally, residual analysis and cross-validation were applied to evaluate the performance of the model. The former plays an important role in validating the regression model, consisting of a test for normality, test for equal variance, test for independence of residuals, and test for spatial autocorrelation, which were developed by ArcGIS and SPSS packages. If the error term satisfies the four basic assumptions of the regression model, then the model is considered valid. The latter was used to assess the accuracy of a linear regression model. In this paper, the leave-one-out cross-validation (LOOCV) method was used to evaluate the model’s performance. Each site was subsequently left out, and a model was developed from the remaining sites with the variables unchanged. Predicted concentrations were estimated and compared with the actual concentrations at each left-out site. The LOOCV *R*^2^ and root mean squared error (RMSE) between the predicted and measured concentrations were calculated for all monitoring stations to represent the model performance.

### 3.3. Population Exposure Assessment Comparisons

#### 3.3.1. Mobile Phone Signal Data

The mobile phone signal dataset in Shanghai, including the time, location, and users’ information for the period from 11 November to 30 November 2019, was provided by JI SMART (http://daas.smartsteps.com/, accessed on 19 January 2021). Initially, the number of mobile phone users included in the data was approximately 5.28 million, and this number can be used to represent Shanghai’s population of approximately 23.55 million, excluding the Chongming district. The main detailed information in the dataset included (1) user attributes, which contained an anonymously processed user ID, gender, age, etc.; (2) date, start time, and end time of stay points; (3) location, which was represented by a grid number with a resolution of 250 m instead of longitude and latitude; (4) grid information containing grid ID, coverage area (well-known text), etc.

The dataset we acquired for the study was processed. More details on the data processing, including the identification of the homes and workplaces of users, were not available due to the confidentiality agreement. Briefly, the locations of each user’s home and workplace were identified according to how long they remained at certain locations. Home locations were recognized based on the location where users remained the longest between 9 p.m. and 8 a.m. during the month. Similarly, users’ workplaces were identified according to the location where they spent the most time between 9 a.m. to 5 p.m. Stay points, excluding the home and workplace, were marked as other activity locations. 

A correlation test and paired samples *t*-tests were used to verify whether there was a significant difference between the seventh population census data and mobile phone signal data, regarding the population distribution, sex composition, and age structure in the districts of Shanghai. It is worth mentioning that population proportion was chosen for comparison rather than the population size due to the different magnitudes of the two sets of data. 

First, population distribution was defined as the proportion of each district’s residential population to the total population of Shanghai, designed to indicate whether the two sets of data were consistent in the geographical distribution of the population. Pearson’s correlation coefficient of population distribution was 0.998, close to 1, indicating that the two samples had a strong linear correlation. The *p*-value from the paired-samples *t*-test was 0.999, which was greater than the significance level of α = 0.05; that is, there was no sufficient evidence to claim that the population distribution from the mobile phone signal dataset differs from the one from the Seventh National Census data as shown in Table 2.

Secondly, the sex ratio defined as the number of females per 1000 males was chosen as a social indicator of sex composition as listed in Table 3. The correlation between the two samples was considered to be strong, because the absolute value of the Pearson’s correlation coefficient was 0.794, greater than 0.75. In addition, the paired sample *t*-test (*p*-value = 0.374) confirmed that there was no statistically significant difference between the seventh population census data and mobile phone signal data. That is to say, the mobile phone signal data could accurately capture the population’s gender structure in the districts of Shanghai.

Thirdly, the proportions of the population aged 0–14, 15–64, and over 65 years were used to describe the population age structure in the districts of Shanghai as listed in Table 4. The correlation test and paired samples *t*-tests were performed separately for each age group, and the Pearson’s correlation coefficients were 0.518, 0.912, and 0.942, respectively, indicating strong positive linear correlations between the three groups of paired samples. Furthermore, the *p*-values of the *t*-tests were 0.997, 0.627, and 0.499, which meant that there was no significant difference in the age structure, suggesting that the mobile phone signal dataset could denote the age structure of different areas in Shanghai.

In conclusion, it was proved that the mobile phone signal dataset used in this study was a representative sample of the population distribution, sex composition, and age structure in Shanghai.

#### 3.3.2. Air Pollution Exposure Assessment

Population-weighted exposure level (PWEL), proposed by Fu and Kan [95], was chosen as the exposure assessment indicator in this paper. It was calculated as predicted concentrations combined with population dynamic distributions. First, NO_2_ pollution during different periods was estimated with the LUR models. Secondly, hourly gridded population data with a spatial resolution of 500 × 500 m was derived from the mobile phone data. Then, using ArcGIS software, the NO_2_ concentration layers were overlaid on the gridded population distribution layers and NO_2_ exposure level in each grid was calculated based on Equation (1) as follows:(1)$$E_{i}=\left(P_{i}\times C_{i}\right)/\overset{n}{\underset{i=1}{\sum }}P_{i}$$
where *i* indicates grids, *n* indicates the number of grids, $E_{i}$ represents grid *i*’s potential population exposure, $P_{i}$ denotes grid *i*’s population size in a certain period, and $C_{i}$ denotes grid *i*’s concentration of specific air pollutants. Moreover, the mean exposure was used as a baseline to classify each grid’s exposure into one of four levels, namely, low risk (<0.5 SD), medium risk (>0.5 and ≤1.5 SD), high risk (>1.5 and ≤2.5 SD), and critical risk (>2.5 SD). Finally, the population-weighted exposure level of NO_2_ for Shanghai was assessed using Equation (2), where E represents Shanghai’s potential NO_2_ exposure:(2)$$E=\overset{n}{\underset{i=1}{\sum }}\left(P_{i}\times C_{i}\right)/\overset{n}{\underset{i=1}{\sum }}P_{i}$$

#### 3.3.3. Population Exposure Comparisons

In this section, NO_2_ exposure characteristics in Shanghai were explored for spatio-temporal variations and group disparity (gender and age), which is crucial for assisting policy-making aimed at attenuating health disparities. Firstly, the diurnal and weekly variations in population exposure were identified based on weekly and daily averages of NO_2_ exposure for the period from 11 to 30 November 2019. Next, NO_2_ exposure levels for each grid area were analyzed to determine spatial variations and to highlight risky areas that required additional attention. Finally, NO_2_ exposure was compared according to gender and age to quantify exposure disparities between different groups.

## 4. Results

### 4.1. Spatio-Temporal Distribution of Pollutant Concentrations

Different influencing factors finally entered the LUR models in different periods. Overall, land use was the most significant indicator, where green spaces and water bodies proved to effectively absorb and purify air pollutants, and then meteorological elements and traffic-related variables as listed in Table 5, Table 6, Table 7 and Table 8. Approximately 56–73% of the variations in NO_2_ pollution were explained by the final models and the RMSEs were 4.554, 4.732, 5.371, and 4.894, respectively, which indicates good performance. The differences between the LOOCV *R*^2^ and the model *R*^2^ were less than 10%, indicating the stability of the LUR model [96]. Moreover, given the z-scores of −0.757, −1.499, −0.842, and −1.295, respectively, the model residuals did not appear to be significantly different from random, fulfilling the regression model assumptions of spatial independence in residuals.

After completing the final models, regression equations were applied to a regular 500 × 500 m grid covering the entire study area. A pollution point map was developed using the predicted values. As shown in Figure 3, an evident spatio-temporal heterogeneity of NO_2_ pollution was observed in Shanghai. Overall spatial variation showed a tendency to decline gradually from downtown in all directions, but the spatial pattern of different periods also presented a uniqueness. Higher NO_2_ values were observed to be transferred from the urban districts with the intensive road network to suburban districts with clustered logistics and industrial parks, such as the Baoshan and Minhang Districts, due to the traffic restriction policy aimed at trucks in Shanghai that encourages freight transport at night. Peak hours shared a similar distribution pattern, but transportation contributed more to the evening peak. In general, NO_2_ concentrations were higher west of than the east of the Huangpu River. In terms of temporal variation, the average values during the daytime, nighttime, morning peak, and evening peak were 29.71, 41.54, 44.07, and 46.52 μg/m^3^, respectively, which exhibited a bimodal diurnal variation, and the figure of the evening peak was higher than the early peak, approximately resembling daily diurnal traffic patterns.

In conclusion, the results of the LUR models suggest that the pollution concentration could be a result in a dynamic game between the air-pollutant emission intensity and environmental self-purification capacity. Consequently, air quality improvements not only require an emphasis on controlling the air-pollutant sources, such as advocating low-carbon travel, prioritizing public transportation development, and advancing emission standards, but also on taking advantage of environmental self-purification capacity. For example, it is necessary to take meteorological elements (wind speed, wind direction, precipitation, etc.) into consideration in the selection of locations for logistics and industrial parks or the height of a chimney. In addition, for air quality improvement, it would be wise to increase the rate of green land use, considering the purification capacity of plants.

### 4.2. Differences in Exposure and Inequality

#### 4.2.1. Temporal Variation

As previously stated, NO_2_ exposure was evaluated by population-weighted exposure levels based on the pollution simulation results from the LUR models and dynamic population distribution from the mobile phone signal data. The mean values, minimum values, maximum values, and standard deviations are summarized in Table 9, presenting bimodal variations consistent with the differences in the pollutant concentration within a day.

The daily mean values of NO_2_ exposure are shown in Figure 4, where the gray shaded areas represent weekends. NO_2_ exposure remained at a high level on weekdays but decreased on weekends. This fluctuation could be attributed to the weekly variations in traffic volumes; that is, there was more traffic on weekdays than on weekends due to the fact of commuters heading to work. However, this trend was not distinct during the nighttime. This was mainly because freight transport did not have an obvious weekly variation. In conclusion, NO_2_ exposure might be influenced by both pollutant concentration and population dynamics.

#### 4.2.2. Spatial Variation

In this section, the standard deviation classification method was used to place the NO_2_ exposure in each grid area at four levels to highlight the differences between the districts. It should be mentioned that there was no comparison between different periods, because this classification was based on the mean value of all air pollution exposure during a single corresponding period.

Higher risk areas were observed in the downtown area with a dense population, as shown in Figure 5, mainly west of the Huangpu River, including the Hongkou, Yangpu, Huangpu, Xuhui, and Jing’an districts. In other words, the areas to the east of the Huangpu River were healthier with lower air pollutant exposure. Furthermore, there was a difference in the range of NO_2_ exposure levels, which were more clustered during the morning peak, indicating exposure inequality was lower during that period. As previously stated, heavy traffic contributed the most to NO_2_ exposure during rush hours. When combined with a large number of cross-regional population dynamics due to the commuting demand, NO_2_ exposure was largely homogeneous across different regions. We would also like to note that NO_2_ exposure inequality during the morning peak was lower, but exposure levels were higher than during other periods. This characteristic was not observed during the evening peak, in part because the flexible off-duty hours led to more scattered commuting.

#### 4.2.3. Group Disparity

To investigate group disparity, user attribute information, including gender and age, was extracted from the mobile phone signal data. The ages were divided into four groups, namely juvenile (≤18), young (19–44), middle-aged (45–64), and elderly (≥65).

Females were subject to more NO_2_ exposure (Figure 6). The exposure disparity between males and females was found to reach a peak during the day, especially during the morning peak, and decreased at night, which could be attributed to gender differences in commuting behavior. In general, females took more trips and had more complex trip chains than men, probably since they undertook a large number of non-work-related trips, such as shopping, delivering children to school, or accompanying the elderly to health centers [97], namely, women had a higher probability of being exposed to traffic-related pollution or other sources of pollution. An inequality in family responsibilities may have evolved into inequality in terms of pollution exposure due to differences in travel behavior. 

As the main bearer of daily housework, women’s destinations of most non-work-related trips were concentrated in urban districts with a completely public transportation system, convenient shopping areas, abundant medical resources, and children’s educational institutions. In a word, urban districts were more attractive to women [98]. However, these places were also heavily polluted areas as described above. Kernel density analysis was applied in ArcGIS to explore the gender differences in travel distribution based on trajectory data extracted from the mobile phone signal dataset, and the standard deviation stretch and gamma correction were applied to the mapping to eliminate the gap in trip volume between different groups and increase the contrast in the raster dataset to highlight the range of hot spots. As shown in Figure 7, the distribution of different gender groups in the road network was both generally of unbalanced and aggregated characteristics, and the travel distribution of women was more clustered. Male trip volume showed a continuous trend of decreasing outward from the urban areas but remained high on some suburban trunk roads. However, there was a very significant cliff-like decline in the trip volume of females in the urban periphery, with fewer trips on major arterial roads in the suburban areas compared to men. It was worth noting that the central area of the city where women gather is precisely the most polluted area, which may lead to the overall higher exposure of women than men.

Regarding the age groups, NO_2_ exposure increased significantly in the elderly as shown in Figure 8; that is, the elderly group endured the greatest exposure, followed by the young and the middle-aged, and then the juvenile groups.

Residential preference and travel behaviors may have contributed to the observed disparity between age groups. Since the 1990s, the policy of suppressing the second industry and developing the third industry has prompted factories in the central area of the city to relocate to the suburbs and gather to form industrial parks. At the same time, the planning and construction of new towns were in full swing under the influence of the Greater Shanghai Plan. The working-age population in the central urban area and from surrounding cities migrated to the suburbs or new towns due to a large number of employment opportunities and moderately priced housing, which contributed to the age structure differentiation between the urban area and the suburban areas of Shanghai. Finally, the elderly population was concentrated in the urban areas while young and middle-aged groups gathered near the suburbs.

According to the seventh Shanghai Census Bulletin, Hongkou and Huangpu districts have the densest elderly population, with a density of more than 5000 people per square kilometer. The elderly population in the suburbs is relatively sparse, such as in Jinshan, Qingpu, and Fengxian Districts, where the density of the elderly population was less than 250 people per square kilometer. Among them, the difference between the densest and the sparsest regions was as high as 35 times. In general, the density of the resident elderly population in Shanghai was low around the middle and uneven in the regional distribution. Furthermore, the characteristics of the circle structure were significant. As listed in Table 10, the population proportion over 60 years in the urban districts of Shanghai exceeded 25%, with an average of 30.2%. However, suburbs were mostly below 20%, with an average of just 19.6%. To this end, the global Moran’s I index was used in ArcGIS to further evaluate the spatial autocorrelation of the people over the age of 65 in the districts of Shanghai. Given the z-score of 3.02, much larger than the critical value of 1.65, there was more than a 95% likelihood that Shanghai’s elderly population could be the result of the clustered pattern. It meant that the spatial distribution of the elderly population has strong homogeneity, that is, it is uneven in the regional distribution.

Based on the residence distribution of different age groups extracted from the mobile phone signal dataset, the spatial distribution of the population density between age groups was described through the mean–standard deviation method in ArcGIS. As shown in Figure 9, the spatial distribution of the elderly population had a more significant clustering characteristic than other age groups. It is worth noting that the urban districts, where the elderly population was densely distributed, were the areas with the most critical NO_2_ pollution. This means that the elderly population with the lowest pollution contribution is bearing the highest risk of exposure; that is, Shanghai has serious environmental inequities in terms of air quality.

## 5. Discussion

### 5.1. Principal Findings

In summary, this study attempted to combine dynamic LUR models with mobile phone signal data to explain the spatio-temporal variations and group disparities of NO_2_ exposure in Shanghai. Our main conclusions can be summarized as follows.

The dynamic LUR models revealed an evident temporal variation of NO_2_ exposure in Shanghai. Overall, it followed a bimodal diurnal variation and remains at a high level on weekdays but decreases on weekends, consistent with changes in traffic volume;In terms of spatial variation, higher-risk locations included urban areas with dense populations and busy traffic and were concentrated west of the Huangpu River. Also, lower regional inequality of NO_2_ exposure was observed during the morning peak due to a large number of cross-regional commutes that led to NO_2_ exposure homogenization across different regions;As for group disparity, women and the elderly proved to suffer more exposure to NO_2_ pollution, which could be attributed to gender differences in travel behavior and the preference of residence in different age groups.

### 5.2. Strengths and Limitations

This study aimed to introduce an improved NO_2_ exposure assessment by integrating the dynamic LUR model with mobile phone signal data and explored the spatio-temporal variation of the population exposure disparity in the case of Shanghai. Nevertheless, some limitations in this study should be pointed out. 

First, limited by the availability of pollution concentration and mobile phone signal data, only diurnal and weekly variations in NO_2_ exposure were explored. In the future, the data could be strengthened and validated on a monthly, seasonal, or annual scale over a longer period. Secondly, this paper attempted to establish a link between traffic-related air pollution and human risk. However, the lack of quantitative analysis of traffic emissions made it difficult to strongly support this conclusion. Thirdly, although this paper has revealed the exposure disparity in terms of spatio-temporal variation and age/gender group, there is still a long way to go in addressing the issues of environmental and health equity. The subsequent research is supposed to further dig out whether the implementation of financial support policies such as subsidies and tax cuts could attenuate exposure inequality and improve public health.

### 5.3. Policy Implications

These results suggest that Shanghai is trapped in an awkward predicament with an uncoordinated relationship between urban development and air quality. Currently, it is necessary to create a comprehensive environmental plan and a target- and results-oriented governance program to improve air quality and public health. 

Based on our analysis, we propose several recommendations. First, air pollution improvement requires regulators to pay close attention to pollution sources. Scientific land-use planning can be an ideal tool to coordinate urban economic development and quality living environments. For instance, it would be prudent to increase the number of green spaces near the pollutant source to prevent air pollutants from spreading elsewhere. In addition, the needs and interests of vulnerable social groups have to be prioritized in the process of the formulation and implementation of urban planning and environmental health policies to ensure environmental justice and sustainable development. Second, transportation improvement programs are also considered an effective way to alleviate the negative impacts of motorization. Today, TOD (transit-oriented development) has become a priority for improving public health and developing a quality living environment, as it encourages enhanced accessibility with a superior and sustainable transit system along with mixed land use and compact urban development. Moreover, the boom in ridesharing and electric vehicles has also created more opportunities to reduce the contribution of the road networks to air pollution. Third, higher risk areas for air pollution exposure are generally downtown in locations with high population density, and population agglomeration aggravates air pollution, as revealed in previous studies [99]. Therefore, the Shanghai municipal government should develop several effective control strategies for population agglomeration to guide the reasonable spatial distribution of the population to reduce the intensity of population exposure inequality.

## 6. Conclusions

In summary, this study proposed a fairly novel approach that integrated a dynamic LUR model and mobile phone signal data to improve the accuracy of dynamic population exposure. A more detailed analysis of group disparity was investigated in terms of spatio-temporal variation in the case of Shanghai. Although some limitations exist, our study could help researchers better understand the spatio-temporal patterns of NO_2_ pollution exposure and assist policy-making aimed at improving public health. The results showed that it is vital to focus on land-use planning, transportation improvement programs, and population agglomeration to attenuate exposure inequality. 

## Figures and Tables

**Figure 1 ijerph-19-05872-f001:**
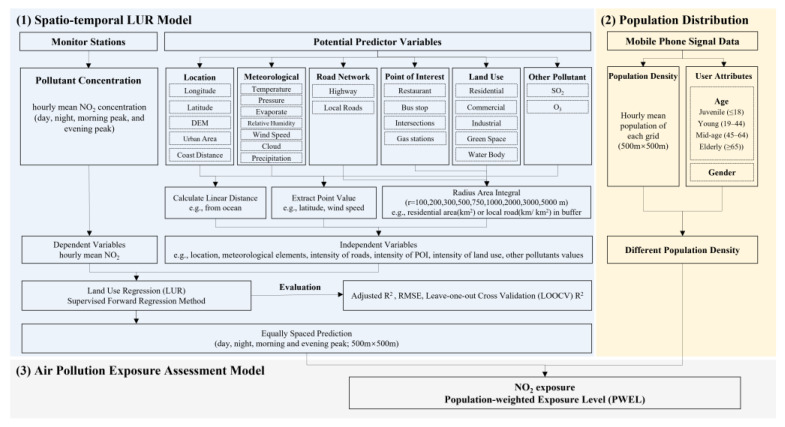
Research flow chart.

**Figure 2 ijerph-19-05872-f002:**
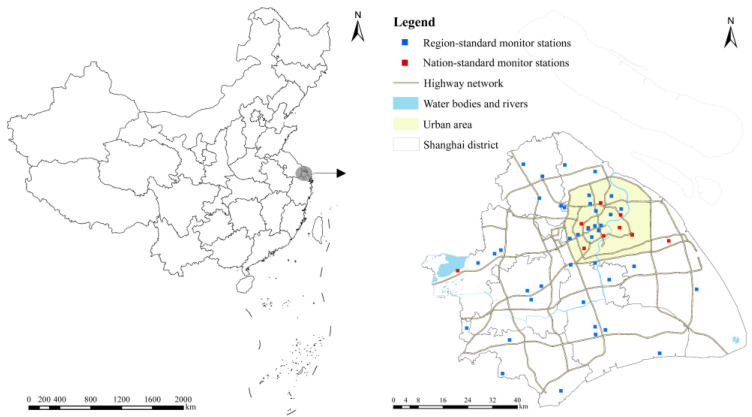
Study area and monitor distributions.

**Figure 3 ijerph-19-05872-f003:**
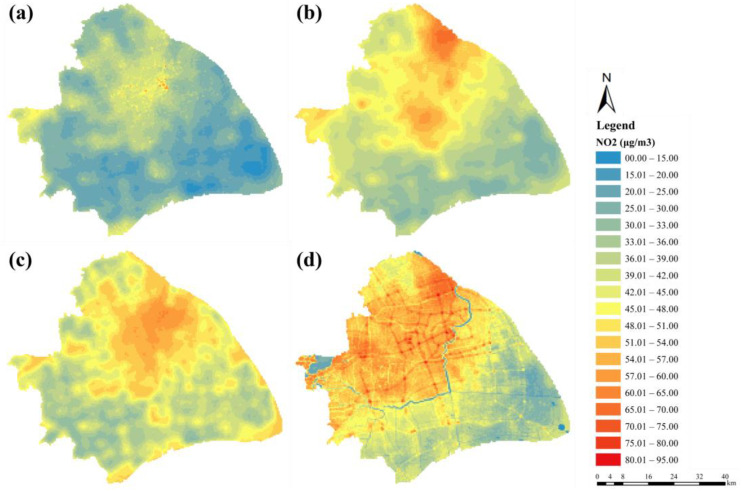
Hourly NO_2_ concentration maps of land-use regression predictions during (**a**) daytime (9 a.m.–5 p.m.), (**b**) nighttime (7 p.m.–7 a.m.), (**c**) morning peak (7 a.m.–9 a.m.), and (**d**) evening peak (5 p.m.–7 p.m.) in Shanghai.

**Figure 4 ijerph-19-05872-f004:**
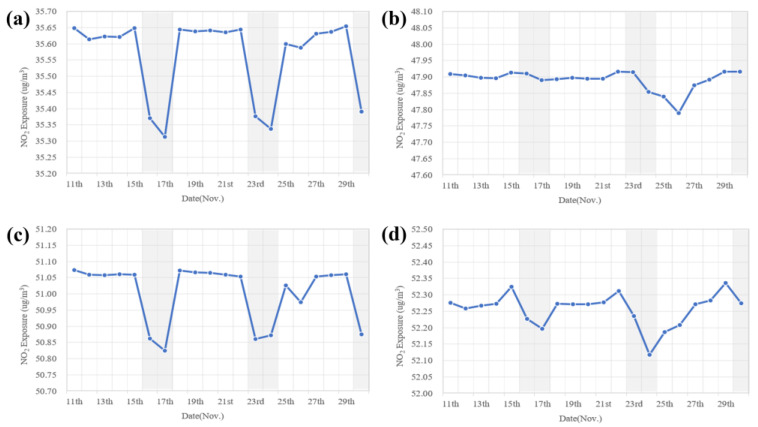
Daily NO_2_ exposure during (**a**) daytime (9 a.m.–5 p.m.), (**b**) nighttime (7 p.m.–7 a.m.), (**c**) morning peak (7 a.m.–9 a.m.), and (**d**) evening peak (5 p.m.–7 p.m.) in Shanghai.

**Figure 5 ijerph-19-05872-f005:**
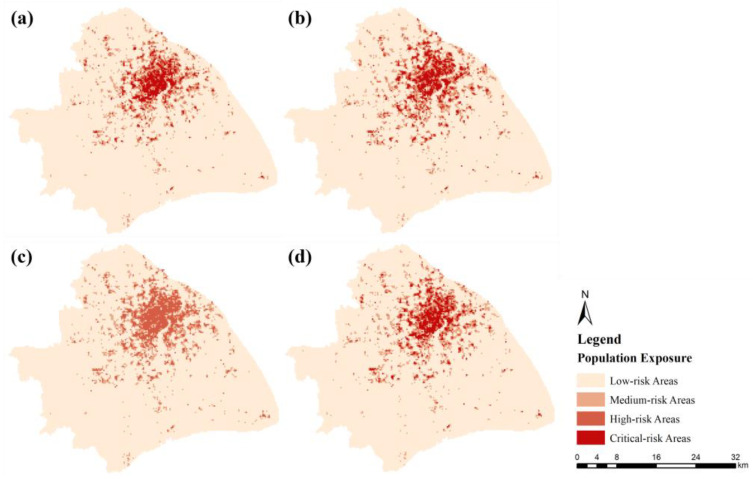
Spatial distribution of NO_2_ exposure during (**a**) daytime (9 a.m.–5 p.m.), (**b**) nighttime (7 p.m.–7 a.m.), (**c**) morning peak (7 a.m.–9 a.m.), and (**d**) evening peak (5 p.m.–7 p.m.) in Shanghai.

**Figure 6 ijerph-19-05872-f006:**
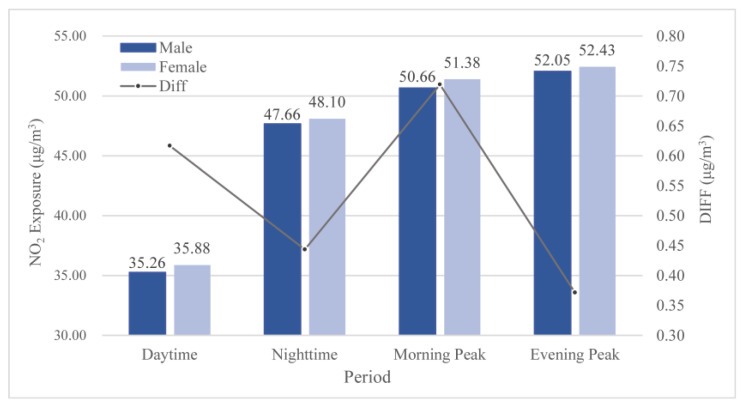
NO_2_ exposure disparity between males and females.

**Figure 7 ijerph-19-05872-f007:**
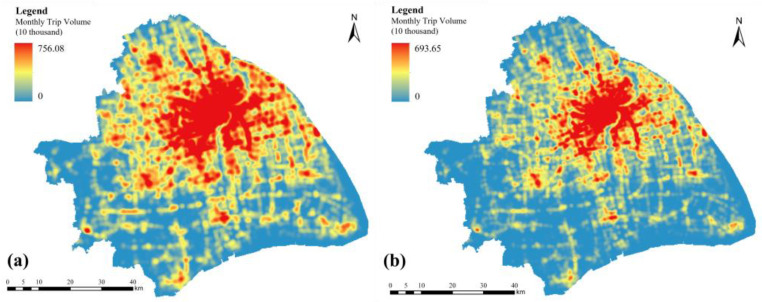
Distribution of travel trajectories for males (**a**) and females (**b**).

**Figure 8 ijerph-19-05872-f008:**
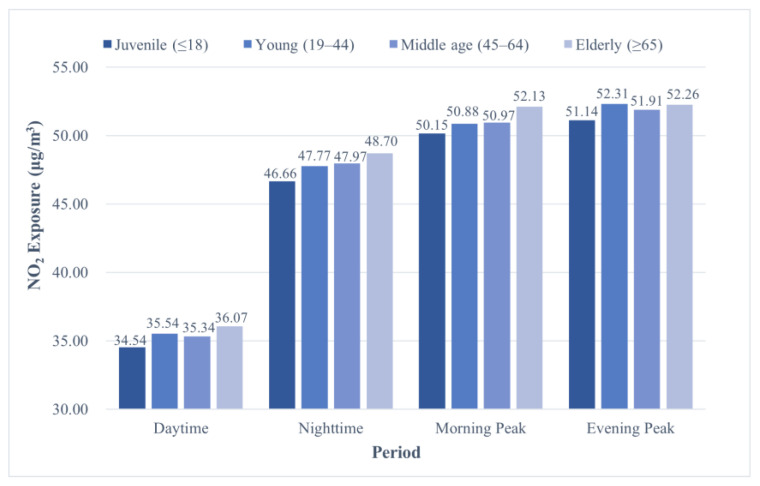
NO_2_ exposure disparity between age groups.

**Figure 9 ijerph-19-05872-f009:**
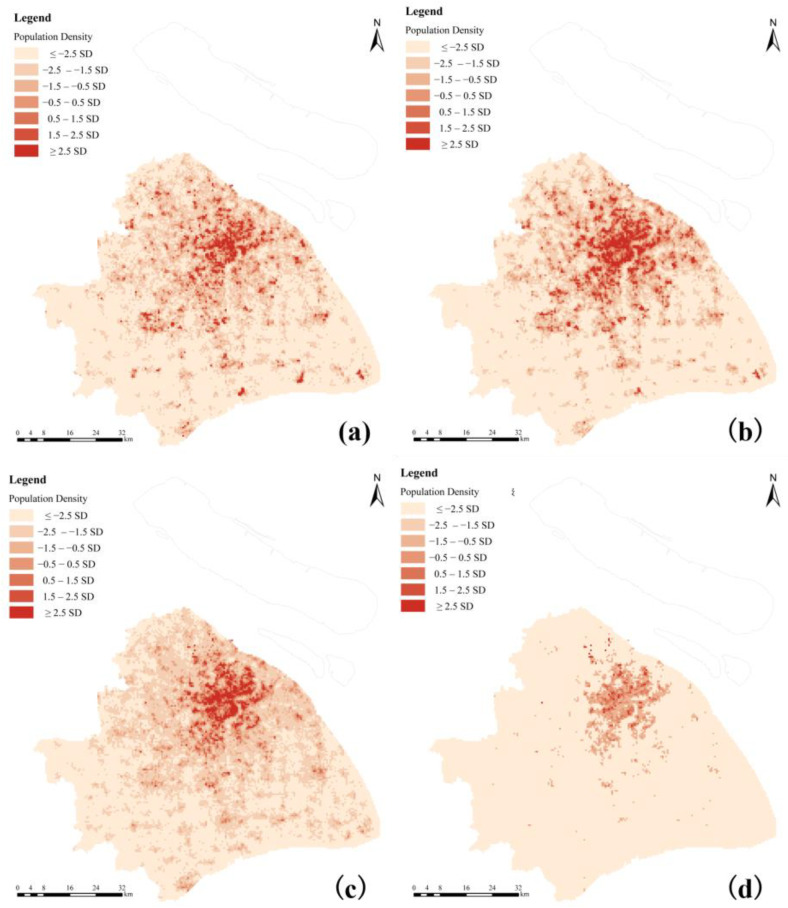
Spatial distribution of residence for (**a**) juvenile (≤18), (**b**) young (19–44), (**c**) middle age (45–64), and (**d**) elderly (≥65).

**Table 1 ijerph-19-05872-t001:** Independent variables.

Category	Variable Sub-Categories	Buffer Radii (m)	Unit	Prior Direction
Location	Longitude	Extract point value /calculate linear distance	°	O
	Latitude		°	O
	DEM		m	−
	Distance to the ocean		km	−
	Suburban and urban area		0/1	+
Meteorological elements	Temperature	Extract point value	℃	−
	Pressure		hPa	+
	Evaporation		mm	O
	Relative humidity		%	O
	Precipitation		mm	−
	Cloud		%	+
	Ground surface temperature		℃	−
	Wind speed		m/s	−
Road networks	Highway	Radius area integral (r = 100, 200, 300, 500, 750, 1000 m)	km/km^2^	+
	Local road		km/km^2^	+
Land use	Residential land use	Radius area integral (r = 100, 200, 300, 500, 750, 1000, 2000, 3000, and 5000 m)	km^2^	+
	Commercial land use		km^2^	+
	Industrial land use		km^2^	+
	Greenspace		km^2^	−
	Waterbody		km^2^	−
Points of interest	Restaurant	Radius area integral (r = 100, 200, 300, 500, 750, 1000, 2000, 3000, 5000 m)	N	+
	Bus stop		N	+
	Intersection		N	+
	Gas station		N	+
Other pollutants	SO_2_	Extract point value	μg/m^3^	+
	O_3_		μg/m^3^	−

Note: “+” denoted assumed positive correlation. “−“ denoted assumed negative correlation, “O” denoted not assigned because no effect could be assumed. More details were described in Section 3.2.3.

**Table 2 ijerph-19-05872-t002:** Population distribution in the districts of Shanghai (%).

District	7th Population Census Data	Mobile Phone Signal Data	District	7th Population Census Data	Mobile Phone Signal Data
Huangpu	2.73	2.14	Baoshan	9.22	9.42
Xuhui	4.59	3.79	Jiading	7.57	8.24
Changning	2.86	3.33	Pudong	23.44	25.66
Jing’an	4.03	3.80	Jinshan	3.40	2.46
Putuo	5.11	5.07	Songjiang	7.88	8.22
Hongkou	3.13	2.53	Qingpu	5.25	5.06
Yangpu	5.13	4.46	Fengxian	4.71	4.72
Minhang	10.95	11.10			

**Table 3 ijerph-19-05872-t003:** Sex ratio in the districts of Shanghai.

District	7th Population Census Data	Mobile Phone Signal Data	District	7th Population Census Data	Mobile Phone Signal Data
Huangpu	107.59	101.63	Baoshan	109.51	100.13
Xuhui	95.06	95.02	Jiading	116.83	114.40
Changning	90.91	98.50	Pudong	108.04	103.32
Jing’an	95.19	96.79	Jinshan	113.74	117.63
Putuo	97.14	101.68	Songjiang	114.70	107.42
Hongkou	97.17	96.43	Qingpu	120.39	120.64
Yangpu	97.91	103.05	Fengxian	116.78	117.26
Minhang	107.72	93.60			

**Table 4 ijerph-19-05872-t004:** Age structure in the districts of Shanghai (%).

District	7th Population Census Data			Mobile Phone Signal Data		
	0–14	15–64	≥65	0–14	15–64	≥65
Huangpu	8.67	72.84	18.48	8.74	73.72	17.54
Xuhui	9.79	69.60	20.61	9.39	71.48	19.13
Changning	8.89	70.46	20.65	9.11	71.00	19.89
Jing’an	9.31	68.72	21.97	10.78	70.20	19.02
Putuo	9.39	69.48	21.12	10.04	69.34	20.62
Hongkou	8.19	68.58	23.23	8.53	69.20	22.27
Yangpu	8.84	69.31	21.85	9.12	70.47	20.41
Minhang	10.86	75.11	14.02	9.57	76.91	13.52
Baoshan	10.01	74.61	15.39	8.54	74.72	16.74
Jiading	9.81	78.12	12.07	9.62	77.39	12.99
Pudong	10.48	74.53	14.99	10.03	76.32	13.65
Jinshan	9.22	73.75	17.03	9.54	70.75	19.71
Songjiang	10.70	78.39	10.91	10.76	76.87	12.37
Qingpu	8.86	79.58	11.56	9.31	76.79	12.90
Fengxian	9.28	76.96	13.76	9.21	77.92	12.87

**Table 5 ijerph-19-05872-t005:** Final results of the NO_2_ model for daytime (9 a.m.–5 p.m.).

Explanatory Variable	β	*t*	*p*	VIF
(Intercept)	2679.989	4.764	0.000	
Green space within 3000 m (km^2^)	−1.212	−6.550	0.000	1.170
Longitude	−22.035	−4.689	0.000	1.643
Intersections within 300 m (N)	0.249	2.656	0.011	1.159
Cloud (%)	0.982	2.529	0.015	1.471
Global statistics	$R_{adj}^{2}$	0.561		
	$\mathrm{LOOCV}\;R^{2}$	0.528		
	RMSE	4.554		

**Table 6 ijerph-19-05872-t006:** Final results of the NO_2_ model for nighttime (7 p.m.–7 a.m.).

Explanatory Variable	β	*t*	*p*	VIF
(Intercept)	227.595	8.500	0.000	
Green space within 3000 m (km^2^)	−0.677	−2.942	0.005	1.775
O_3_ (μg/m^3^)	−0.476	−3.342	0.002	1.595
Relative humidity (%)	−1.697	−6.164	0.000	2.168
Wind speed (m/s)	−5.259	−3.000	0.004	2.058
Bus stops within 2000 m (N)	0.039	3.281	0.002	1.894
Distance to the ocean (km)	0.159	2.060	0.045	1.988
Global statistics	$R_{adj}^{2}$	0.692		
	$\mathrm{LOOCV}\;R^{2}$	0.650		
	RMSE	4.732		

**Table 7 ijerph-19-05872-t007:** Final results of the NO_2_ model for the morning peak (7 a.m.–9 a.m.).

Explanatory Variable	β	*t*	*p*	VIF
(Intercept)	52.481	22.336	0.000	
Green space within 2000 m (km^2^)	−2.318	−4.171	0.000	1.736
Gas stations within 5000 m (N)	0.124	2.480	0.017	1.736
Global statistics	$R_{adj}^{2}$	0.560		
Wind speed (m/s)	$\mathrm{LOOCV}\;R^{2}$	0.528		
Bus stops within 2000 m (N)	RMSE	5.371		

**Table 8 ijerph-19-05872-t008:** Final results of the NO_2_ model for the evening peak (5 p.m.–7 p.m.).

Explanatory Variable	β	*t*	*p*	VIF
(Intercept)	213.666	8.481	0.000	
O_3_ (μg/m^3^)	−0.251	−2.013	0.051	1.902
Green space within 2000 m (km^2^)	−1.105	−2.349	0.024	1.569
Relative humidity (%)	−2.077	−5.711	0.000	1.405
Distance to the ocean (km)	0.379	4.128	0.000	2.574
Water body within 300 m (km^2^)	−145.268	−3.215	0.003	2.206
Highway intensity within 500 m (km/km^2^)	1.427	2.093	0.042	1.192
Intersections within 300 m (N)	0.212	2.027	0.049	1.306
Global statistics	$R_{adj}^{2}$		0.737	
	$\mathrm{LOOCV}\;R^{2}$		0.705	
	RMSE		4.894	

**Table 9 ijerph-19-05872-t009:** Summary statistics of NO_2_ exposure (μg/m^3^).

Period	Mean	Minimum	Maximum	Standard Deviation
Daytime	35.563	35.226	35.695	0.127
Nighttime	47.891	47.693	48.027	0.061
Morning Peak	51.005	50.802	51.146	0.107
Evening Peak	52.257	52.112	52.337	0.050

**Table 10 ijerph-19-05872-t010:** Population Aging Level in the districts of Shanghai.

Location	District	Population Proportion over 60 (%)	Population Proportion over 65 (%)	Population Density over 65 (/km^2^)
Urban District	Huangpu	26.6	18.5	5963.6
	Xuhui	28.7	20.6	4176.0
	Changning	29.1	20.7	3737.1
	Jing’an	31.6	22.0	5794.0
	Putuo	30.6	21.1	4716.4
	Hongkou	33.2	23.2	7504.7
	Yangpu	31.8	21.9	4480.4
Suburban District	Minhang	19.5	14.0	998.8
	Baoshan	22.9	15.4	941.5
	Jiading	17.8	12.1	476.8
	Pudong	21.6	15.0	703.7
	Jinshan	23.6	17.0	228.6
	Songjiang	15.8	10.9	344.6
	Qingpu	16.6	11.6	217.4
	Fengxian	19.4	13.8	214.1

## Data Availability

The data presented in this study are available on request from the corresponding author. The data are not publicly available due to the privacy restrictions.

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
