# Peer review of "Spatio-Temporal Variation-Induced Group Disparity of Intra-Urban NO_2_ Exposure"

_ijerph, 2022, doi:10.3390/ijerph19105872_

Round 1

Reviewer 1 Report

A very well written paper, looking into an interesting subject. Information flows nicely through the paper and every claim is backed by proper references. It was an exciting case of extracting demographic data from the mobile phone network. I read the entire paper and the following are my suggestions for further improvement of the paper:

In the version that I had, all figures were low resolution and it was sometimes hard to read the details from the figure. If that is going to be the case, also for the main paper, I recommend you replace them with higher resolution figures.

Line 157: Considering that is repeated.

Line 202 – 206: Does the order at which the parameters are introduced into the model matter? If not, please mention it.

Line 245-253: Can you explain this part further? It is not clear to me what is correlated against what? You can put some of the most important correlation graphs in the supplement information section. These P values do not seem reasonable to me, considering the embedded uncertainties of census data.

Line 280: From what I perceived, you assume the population does not leave the grid cell for that duration. If right, please mention it.

Figure 3: add more explanation in the caption please. It is not clear if these values are total emissions/pollutions or the average for the time period. Also, please mention the differences in the length of each time period.

Line 398: Correct the writing please.

Line 446: It is not clear what you are trying to say. Please consider rewriting.

Reviewer 2 Report

This paper investigates the “Spatio-temporal Variation Induced Group Disparity of Intra-urban NO2 Exposure”.

General comments:

  1. The resolution of the figures needs to be improved.
  2. “Considering that Considering that” in line 157 should read “Considering that”
  3. “higher levels of NO2 than whites” in line 122 should read “NO2
  4. What is the Pi shown in Eq (1) & (2)?
  5. Please provide the references for each equation that you adopt in this article.

Major comments:

  1. Please explain more in detail why select the timespan from November 11th to November 30th, 2019.
  2. How does NO2 affect humans and their health risks? Please discuss more in the first two sections.
  3. Traffic-induced NO2 could be influenced the health risk but that is not the only factor. How about the other contributions such as industrial activity, power plants, and off-road equipment in Shanghai? How about their proportions to NO2 otherwise it is hard to say that the only influence belongs to traffic.
  4. In this work, traffic activity exerts a domain control that influences human risk. However, the author did not provide any traffic data as well as their diurnal cycle which is hard to verify the conclusions.
  5. Conclusions that are too lengthy often have unnecessary. It should be completely rewritten.

Reviewer 3 Report

Thank you for giving me this opportunity to read the manuscript entitled "Spatio-temporal Variation Induced Group Disparity of Intra-urban NO2 Exposure". The topic of this manuscript is interesting and would be a good contribution to this field. I think it could be considered for publication in International Journal of Environmental Research and Public Health once the following issues are addressed.

  1. Please replace the keywords that already appear in the manuscript’s title with close synonyms or other keywords, which will also facilitate your paper to be searched by potential readers.

  1. Lines 87-90: “…, as it ignores variability in exposure caused by individual mobility [44-49]…”. Another newly published papers titiled “Dynamic assessments of population exposure to urban greenspace using multi-source big data” are suggested to be cited here to support the statement here. Besides, social media data are also very widely used dataset used to represent human mobility in environmental exposure assessment. Therefore, some more social media-based references should be added in Lines 88-90, for example, the paper titled “Dynamic assessment of PM2. 5 exposure and health risk using remote sensing and geo-spatial big data”

  1. Please improve the resolution of the Figures used in the manuscript, as the current pictures are not clear and it is hard to read the text information.

  1. Figure 2: The boundaries of China's map are completely wrong. In addition to the absence of information on the nine-dashed line, the western border of Xinjiang is also wrong. Also, please use the appropriate projection for projecting the map of China.

  1. Lines 105-106: “Environmental justice believes that, regardless of the socio-economic status, all residents ought to shoulder equal adverse health impacts from deteriorating environmental conditions [63].” I suggest the authors add some newly published paper to support the statements here, for example, the paper titled “Observed inequality in urban greenspace exposure in China”.

  1. A Limitation section is suggested to be added in the Discussion as a sub-section.

  1. Some grammatical errors exist in the manuscript. Therefore, a critical review of the manuscript language will improve readability.

Round 2

Reviewer 2 Report

The manuscript can now be considered for acceptance, particularly that the authors have addressed the comments in their revision. If the other reviews are satisfied with the current revision, then the paper can be considered for publication

Reviewer 3 Report

Thank you for giving me this opportunity to read the revised version of the manuscript titled "Spatio-temporal Variation Induced Group Disparity of Intra-urban NO2 Exposure", and for the detailed responses to my earlier comments. I am satisfied with this revised version, and I think it is acceptable now.